# Learning Facial Motion Representation with a Lightweight Encoder for Identity Verification

Zheng Sun ⓘ, Andrew W. Sumsion ⓘ, Shad A. Torrie ⓘ and Dah-Jye Lee *ⓘ

Department of Electrical and Computer Engineering, Brigham Young University, Provo, UT 84602, USA; zsun2@student.byu.edu (Z.S.); andreww9@byu.edu (A.W.S.); shad.torrie@byu.edu (S.A.T.)
* Correspondence: djlee@byu.edu

**Abstract:** Deep learning became an important image classification and object detection technique more than a decade ago. It has since achieved human-like performance for many computer vision tasks. Some of them involve the analysis of human face for applications like facial recognition, expression recognition, and facial landmark detection. In recent years, researchers have generated and made publicly available many valuable datasets that allow for the development of more accurate and robust models for these important tasks. Exploiting the information contained inside these pretrained deep structures could open the door to many new applications and provide a quick path to their success. This research focuses on a unique application that analyzes short facial motion video for identity verification. Our proposed solution leverages the rich information in those deep structures to provide accurate face representation for facial motion analysis. We have developed two strategies to employ the information contained in the existing models for image-based face analysis to learn the facial motion representations for our application. Combining with those pretrained spatial feature extractors for face-related analyses, our customized sequence encoder is capable of generating accurate facial motion embedding for identity verification application. The experimental results show that the facial geometry information from those feature extractors is valuable and helps our model achieve an impressive average precision of 98.8% for identity verification using facial motion.

**Keywords:** facial motion encoding; contrastive learning; sequence learning; deep learning

## 1. Introduction

Human face analysis is one of many important computer vision tasks. With the rapid development of consumer electronics, high-resolution camera has become an affordable and powerful input device for applications in our daily life. With this trend comes potential applications including gaining access to a restrictive facility with our face [1], creating our own animated emoji [2], and even remotely fitting new eyeglasses [3], to name a few. All these applications need computer vision algorithms specifically designed for analyzing human face.

In the past decade, deep learning has enabled machines to understand the content in images significantly better than ever before. This revolution in image analysis ensures the development of more accurate face detection, efficient face representations, and realistic face rendering. Even with these tremendous achievements, deep learning is still facing challenges such as neural networks that are easily spoofed [4], models powered by neural networks that are computationally expensive and not suitable for real-time embedded applications, and the granularity of facial motion representation are coarse and not adequate for practical use.

To address the aforementioned challenges, we propose a solution for extracting efficient facial motion embedding (FME) from video for facial motion based applications. Many computer vision applications require facial motion information. Figure 1 shows an example of how facial motion analysis improves the security of an identity verification system [5]. This system processes facial appearance and facial motion currently to

perform identity verification. Our research focuses on facial motion analysis to measure the similarity between two facial motions, one from the enrollment and one from the verification process.

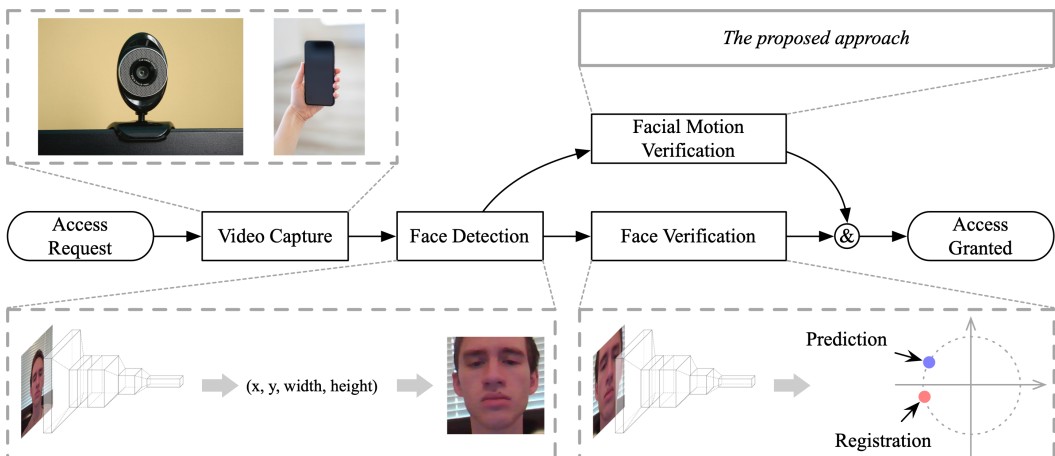

**Figure 1.** Facial motion augmented identity verification system. The shown face is one of the subjects in our dataset introduced in Section 2.4.

Instead of using a limited number of predefined facial expressions like [6], a good face verification system should allow the users to make their own customized facial motion and use it as a secret password. This added security inevitably increases the complexity of the network model. Many deep neural networks for computer vision tasks are computationally expensive and require dedicated hardware such as graphics processing unit (GPU) or application-specific integrated circuit (ASIC) to accelerate the computation for real-time applications. Processing speed is even more challenging for video. An accurate and efficient facial motion encoder is essential to video-based facial motion analysis.

### 1.1. Background

Although this research on representation learning for customized facial motions is new, we have found related research works in the literature that solved a subset of this problem, or proposed essential approaches which help and inspire the development of our efficient facial motion encoder. We discuss research works related to our work in this section.

### 1.1.1. Video-Based Facial Expression Recognition

The most related research topic to our work is video-based facial expression recognition (FER). With the development of deep learning technologies, the-state-of-art performance on the CK+ dataset [7], a popular benchmark on video-based facial expression recognition, reached an impressive 99.69% overall accuracy [8]. Comparing to the image-based FER approaches [9,10], which use deep convolutional neural networks (CNNs) extensively to train a classifier, video-based FER use specific neural network architectures to extract the temporal information across frames. For example, [11–13] used 3D CNN in their models, and [8,14] both adopted the attention mechanism, which was initially designed for machine translation task [15]. Similar to facial expression recognition, other tasks such as micro-expression recognition [16], implicit preference estimation [17], and facial nerve paralysis evaluation [18], also take advantage of short face videos and run facial motion analysis over consecutive frames.

The main objective of video-based FER is developing an accurate classifier to classify a small number of predefined facial expressions in video. It is unrealistic to take a similar approach to recognize random or customized facial motions for our application. This is because there is no unified descriptions of these freestyle facial movements, and the same facial motion made by two individuals could appear different. However, it is possible to

design a specialized network to measure the similarity between two facial motions for our identity verification application. In this work, we develop an efficient facial motion encoder and adopt contrastive learning to train it.

1.1.2. Pre-Training Approaches

Transfer learning is a popular technique for face analysis. Bulat et al. [19] evaluated the impact of both supervised and unsupervised pretraining methods on face-related tasks with static face image input. In their conclusion, transfer learning did not have any negative impact on the downstream face analysis tasks, and in some cases, it actually improved the performance. That means when investigating a new subtopic in human face analysis, it is not always necessary to collect a large and curated dataset. Instead, the robust pretrained neural networks can be helpful for the subsequent vision tasks. Guan [20] adopted transfer learning in their micro-expression spotting task. Initially, they trained a ResNet model using facial expression and emotion datasets. This step provided a spatial feature extractor (SFE) that projects each frame into a vector. Then, they stacked the final feature vectors for all frames into a sequence and used the sequential features as the recurrent neural network input. This approach is similar to our strategy for developing a robust facial motion analysis model.

1.1.3. Lightweight Structures for Face Analysis

Many computer vision applications are hosted on low-power devices like smartphones. To meet the requirements for these embedded application, researchers have designed lightweight architectures such as ShuffleNet [21] and MobileNetV2 [22]. They require only a remarkable one tenth of parameters of the ResNet series while maintaining comparable accuracy for image classification tasks. The global average pooling in these lightweight architectures however, affects the performance on face recognition tasks [23]. They proposed a new design that keeps the residual bottleneck from MobileNetV2, but replaces the global average pooling with a global depthwise convolution layer to improve performance. The final curated model, MobileFaceNet, achieved 99.55% accuracy on face verification tasks with the LFW dataset [24], while the state-of-the-art model achieved 99.83% with much higher computation cost.

Considering the progress achieved in the research works discussed above, we propose an efficient encoder to obtain accurate representation of customized facial motions in real time. Inside this neural network based encoder, we use MobileFaceNet as the spatial feature extractor to extract the geometry information in static frames and a transformer based sequence encoder to obtain the temporal information across multiple frames.

## 2. Methods

The proposed facial motion encoder uses a custom neural network model to extract an accurate representation of facial motion. Figure 2 shows the main idea of our design. There are two modules in our model, a spatial feature extractor and a sequence encoder.

The first step of our method is face detection using BlazeFace [25], a lightweight face detector developed by researchers at Google. The inference of BlazeFace using an embedded device is much shorter than inference of comparable models. For example utilizing the GPU on the Apple iPhone XS , Blazeface inference time is 0.6 ms compared to MobileNetV2-SSD's inference time of 2.1 ms. We perform face detection on every frame in the video even though the subject's face location during recording only changes subtly. This is to minimize unnecessary variations in face image for face feature extraction because the spatial feature extractor is pretrained on face regions cropped by a face detector.

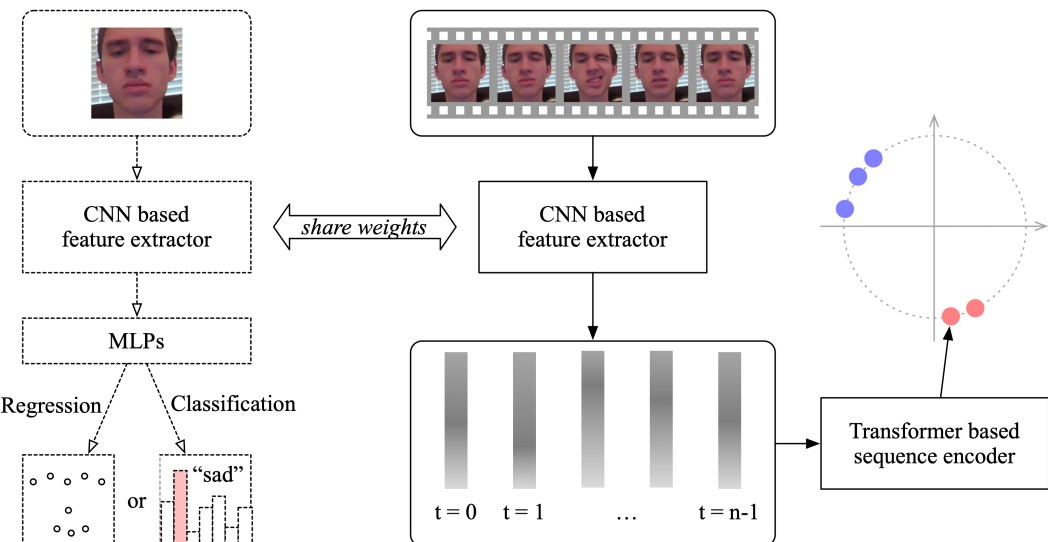

**Figure 2.** Network design.

### 2.1. Spatial Feature Extractors

Our spatial feature extractor is a partition of the CNN model that initially trained for image-based face analysis (as shown on the left of Figure 2) such as FER or facial landmark detection (FLD). Our hypothesis is that, an efficient facial expression classifier or facial landmark detector must pay attention to facial action on the face. By tuning this CNN model on a large-scale curated dataset, we can build an efficient spatial feature extractor, which is important to facial motion representation learning. To avoid the high computation cost with CNN models like VGG-16 or ResNet series, we choose MobileFaceNet that can achieve real-time performance using a modern CPU and obtain comparable accuracy with the state-of-the-art networks.

### 2.2. Sequence Encoder

The sequence encoder stacks features extracted by MobileFaceNet from all frames in the same order as how they appear in the video clip. With these sequential features, our transformer based sequence encoder predicts a normalized feature vector as the embedding of the facial motion in the video.

Figure 3 shows the design of our sequence encoder. For the input sequential features, it calculates the positional encoding at first, adds this encoding to the input sequence, then feeds the position-coded sequence to the stacked transformer layers. Each transformer layer includes a self-attention module and a feed-forward layer. The output of the transformer structure has the same shape as input, $L \times m$, where $L$ is the sequence length and $m$ is the vector space dimension determined by the output layer size of MobileFaceNet. Instead of only paying attention to the feature vector at each sequence end, our encoder selects the maximum value of elements alongside the time axis. As the input sequence of the stacked transformer layers is position-coded, the output of this max-pooling operation still preserves the ordering information of sequence. Then, these variable-length outputs all collapse to a vector in $\mathbb{R}^m$. The final linear layer in sequence encoder projects the vector from $\mathbb{R}^m$ to $\mathbb{R}^n$, where $n$ is the target size of facial motion embedding.

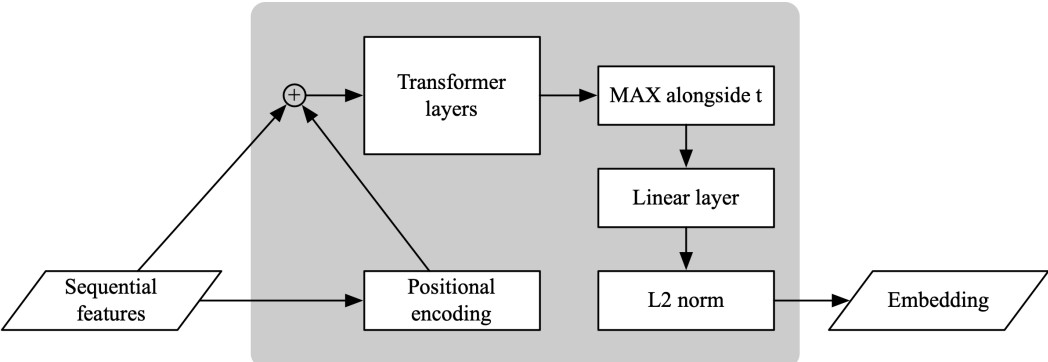

**Figure 3.** Sequence encoder.

*2.3. Loss Functions*

We use contrastive learning during the training process and assemble the positive and negative video pairs inside the mini-batch. Each pair contains two video clips, the positive pairs include two videos with the same facial motion, while the negative pairs are from different motions. To avoid the interference from subject variations, both positive and negative pairs are formed using the videos from the same subject.

We implement contrastive loss [26] and N-pair loss [27,28] functions. Both rely on pairwise similarity. Our contrastive loss takes the following form.

$$\mathcal{L}^c(i,j) = (1 - y_{i,j})(1 - S_W(i,j)) + y_{i,j}max(0, \ S_W(i,j) - m_c) \tag{1}$$

The total loss of a mini-batch is

$$\mathcal{L}^c_{total} = \frac{1}{N}\sum_i \frac{1}{N-1}\sum_{j\neq i}\mathcal{L}^c(i,j) \tag{2}$$

If two facial motion embedding vectors are from the same facial motion, $y_{i,j}$ equals to 0, otherwise, it is 1. $m_c$ defines the similarity boundary between two different motions. $S_W(i,j)$ is the cosine similarity that is determined by two motion embedding vectors predicted by model with parameter set $W$ as

$$S_W(i,j) = cos\theta = \frac{\vec{v_{i,W}} \cdot \vec{v_{j,W}}}{\|\vec{v_{i,W}}\|\|\vec{v_{j,W}}\|} \tag{3}$$

Because the output embedding of our model is normalized. Equation (3) can be simplified to

$$S_W(i,j) = \vec{v_{i,W}} \cdot \vec{v_{j,W}} \tag{4}$$

Another loss function we employ is the N-pair loss as shown in Equation (5). It is based on cross-entropy loss and is tailored for deep metric learning.

$$\mathcal{L}^{np}(i,j) = -\log\frac{\exp(S_W(i,j)/\tau)}{\sum_{k\neq i}\exp(S_W(i,k)/\tau)} \tag{5}$$

The denominator in Equation (5) is the summation of the exponential similarities of all $N-1$ pairs formed with $\vec{v_i}$. $\tau$ is a scale factor or temperature parameter. We calculate this loss only on positive pairs. The total loss of a mini-batch is

$$\mathcal{L}^{np}_{total} = \frac{1}{N}\sum_i \frac{1}{|P|}\sum_{j\in P}\mathcal{L}^{np}(i,j), \tag{6}$$

where $P = \{j < N \mid j \neq i, \ y_{i,j} = 0\}$.

### 2.4. Dataset

We train the encoder using contrastive learning approach, which relies on both positive and negative sample pairs to calculate the losses. For this facial motion representation study, it is much easier to collect negative sample pairs (clips with different facial motions) than positive pairs. Positive pairs require the same customized facial motion from the same subject to be recorded multiple times, whereas any two facial motions made by different subjects can be used as negative pairs.

Recent self-supervised learning research work [29–31] concluded that positive pairs are not required for some image processing tasks. This conclusion does not apply to video understanding tasks such as facial motion analysis. During self-supervised learning for those image processing tasks, positive image pairs are generated by applying image transforms to a single image. These transforms usually do not change the semantic content in the image. The transformed images are considered as identical or similar to the original image. In contrast, our facial motion representation research uses video clips. Using video clips creates a higher level of complexity beyond simply applying transforms to images. It involves temporal variation and destabilization from frame to frame besides variations in semantic content in individual frames.

Our research on facial motion analysis for identity verification is quite unique. Although it is possible to fabricate video clips with image transforms, they would not include temporal information and destabilization that are found in the recorded videos. Generating realistic positive video pairs of unique facial motions is outside the scope of this research. Collecting sufficient positive video pairs for our research is quite challenging which limits the number of facial motions and subjects in our dataset.

All benchmark datasets for research on facial motion analysis were created for applications such as facial expression or emotion recognition. They either do not include positive video pairs or include only video clips with just a handful of predefined facial expressions or emotions. They were not created for measuring similarity between two random or user-selected facial motions. Our customized dataset includes videos of user-selected facial motions and lip movements. Each subject records five of their chosen unique facial motions and five lip movements that are made based on their own chosen words. Each facial motion and lip movement are repeated ten times. The collected videos were then reviewed to exclude motions with tongue movements or micro-expressions that do not match the scope of this research. Our current dataset contains 59 subjects and a total of 5411 videos. Figure 4 shows four examples of user-selected facial motions and lip movements.

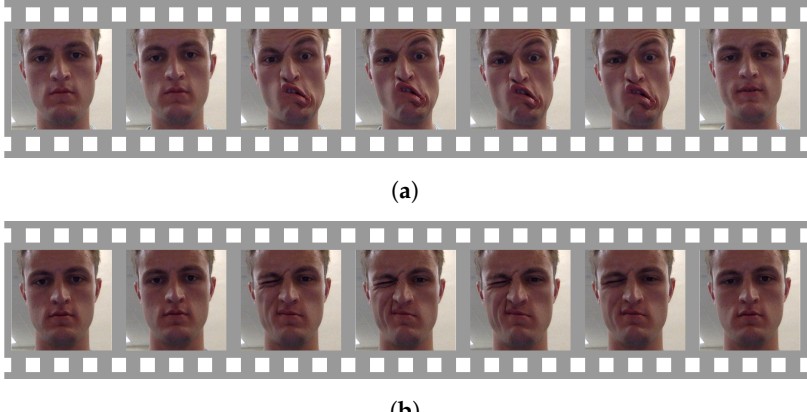

(a)

(b)

**Figure 4.** *Cont.*

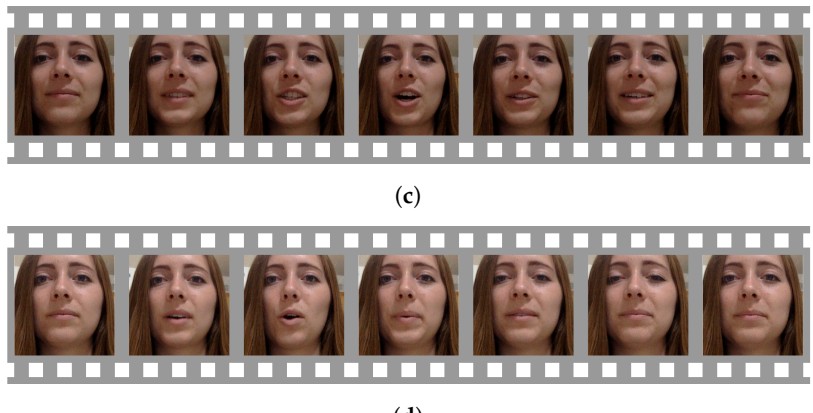

(c)

(d)

**Figure 4.** Examples of our facial motion analysis dataset. (**a**,**b**) are customized facial motions while (**c**,**d**) are lip movements.

### 2.5. Augmentation

Image transforms like cropping, resizing and color distortion for data augmentation are typically helpful during training for face image and other static image analyses. These transforms should only be applied when tuning spatial feature extractor for the upstream tasks. Since the output of the robust spatial feature extractor is remarkably consistent, no data augmentation to the video frames is needed during training.

However, it is logical to create a new clip by randomly removing the non-active frames with neural face at sequence's two ends, when we know the position of onset and offset frames. We name this augmentation method random trimming. The onset and offset of the facial motion are detected using the results from a 68-points facial landmark detector [32]. The standard deviations of the coordinates of these landmark points at different window sizes are calculated to determine the number of frames with neural face before the onset and after the offset of the facial motion.

To demonstrate the robustness of our encoder, in evaluation phase, we extract the facial motion embedding with the full video clips without motion spotting in advance.

## 3. Experiments and Results

We used ten-fold cross-validation to evaluate the performance of our model for face motion verification under different experimental settings. Each experiment contained ten iterations. Nine folds were used for training and the remaining one fold was isolated for evaluation during each loop.

### 3.1. Metrics

In the evaluation phase, we computed the pairwise distances for all possible combinations of positive and negative facial motion pairs in the test set. The precision-recall (PR) analysis was performed on these pairwise distances and labels. Two metrics, average precision (AP) and peak $F_1$-score were used to measure the encoder's performance.

### 3.2. Customized Facial Motions

The first experiment focused on the customized facial motions included in our dataset without the lip movement videos. We compared two spatial feature extractors, one from the facial expression recognition model (SFE-E), the other from the facial landmark detection model (SFE-L). Both feature extractors use the MobileFaceNet architecture to extract facial features. They have identical network layout but with different weights. The SFE-E is tuned with the RAF-DB dataset [33] with 7 emotion categories. Meanwhile, SFE-L is pretrained on the 300-W dataset [34] for generating 68 facial landmark points.

Table 1 shows our encoder's performance for facial motion verification using two feature extractors. SFE-L performed approximately 2% and 4% better than SFE-E in terms of average precision and peak $F_1$-score, respectively. This means the FLD network is better

suited for upstream static face analysis and contains a better spatial feature extractor for our facial motion encoding task than the FER network. This result agrees with our assumption that the 68-point facial landmark detector considers the whole face region, while the facial expression classifier only focuses on the regions related to a small number of predefined facial expressions.

Table 1 also includes the ablation study result with random trimming augmentation method. The result shows that, random trimming only boosted the performance slightly but did improve the performance of the model. For all above combinations, the contrastive loss performed slightly better than the N-pair loss.

**Table 1.** Performance of facial motion verification with different experimental settings.

| Feature Extractors | Loss Functions | Random Trimming | Average Precision | Peak $F_1$-Score |
|---|---|---|---|---|
| SFE-E | contrastive | Y | 0.967 | 0.914 |
| | contrastive | N | 0.966 | 0.912 |
| | N-pair | Y | 0.969 | 0.918 |
| | N-pair | N | 0.969 | 0.914 |
| SFE-L | contrastive | Y | **0.988** | **0.959** |
| | contrastive | N | 0.987 | 0.955 |
| | N-pair | Y | 0.986 | 0.957 |
| | N-pair | N | 0.986 | 0.955 |

*3.3. Lip Movements*

We also included both the facial motion (FM) and lip movement (LM) data in our experiments to evaluate the encoder's performance with lip movements. The results are shown in Table 2.

**Table 2.** Performance study on lip movements.

| Training Data | Test Data | Average Precision | Peak $F_1$-Score |
|---|---|---|---|
| LM | LM | 0.819 | 0.742 |
| FM, LM | LM | 0.816 | 0.741 |
| FM, LM | FM | 0.984 | 0.951 |

When the training data and test data both had only the lip movements, the AP is 81.9% and lower than the 98.8% as shown in Table 1 with only the facial motion data for training and testing. The performance on FM data also dipped slightly from 98.8% to 98.4% when we added the lip movement data to the training set. Although the decrease is almost unnoticeable, it is an indication that the LM data affects the convergence of our model for facial motion verification, and the facial motion encoder is not able to provide accurate description of all types of lip movements, especially for those move subtly.

Another error source is the insufficient data. In our dataset, we group lip movements using the corresponding spoken words, which tie to the phonemes. However, the image and video data only record visemes or lip shape information. As several phonemes could correspond to a single viseme, our pair labels in training are not highly accurate and hence hinder the convergence of our model. Lee et al. demonstrated that lip shape information is insufficient and the tongue position information provided by other sensors is essential to the lip movement representation study [35].

## 4. Discussion

Based on the results presented in Tables 1 and 2, we were able to draw a conclusion that the pretrained spatial feature extractor for facial landmark detection task and our random trimming augmentation method help the training of a good facial motion encoder. Even though our efficient encoder achieved an impressive average precision of 98.8% on

customized facial motions, it would be beneficial to know what the failed cases are and how to improve our model in our future research.

### 4.1. Failed Cases

We studied the failed (false positive and false negative) cases during the testing. These cases provided valuable information about the weakness of our facial motion encoder, and proved the feasibility of our approach from a different perspective.

#### 4.1.1. False Positive Cases

When a video pair contains different motions, their pairwise distance should be larger than the set threshold and fail identity verification. Those few false positive cases in our testing reported a small pairwise distance, or were mistakenly considered similar. This mostly happened in cases that have flash motions, facial motions that occur only in a very few frames and most frames (>70%) in the video present a neutral face. When both videos in a negative pair have flash motion, our encoder will generate two similar representations for them and result in a short pairwise distance or false positive case.

One example of false positive pair is shown in Figure 5. The first (left) and the last three frames for both videos are with the neutral face. These neutral face frames are almost identical. The difference between the two videos is from Frames 2 to 4. The subject raises his eyebrows in Figure 5a. He winks his left eye in Figure 5b. The subtle difference sometimes causes the encoder to generate representations that have a short pairwise distance. We currently use and prefer to use the whole recorded videos for evaluation in order to make it a user-friendly identity verification process. The flash-motion issue could be addressed by applying motion spotting to detect the active frames before sending an image sequence to the encoder.

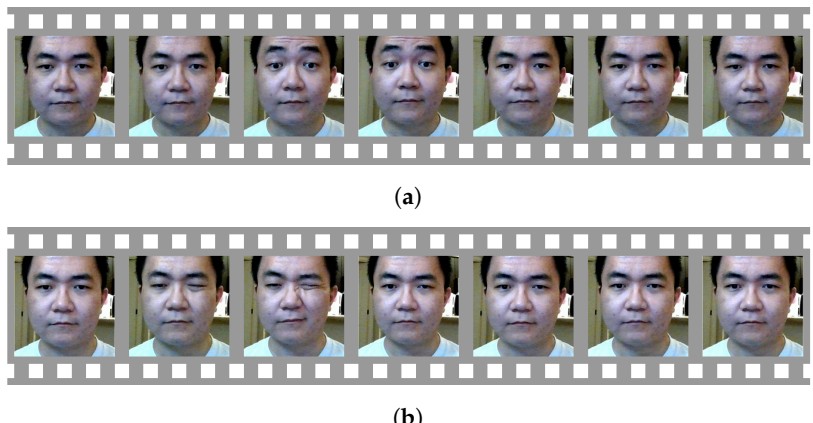

(a)

(b)

**Figure 5.** False positive example. (**a**) is a eyebrow-raising motion. (**b**) is a wink motion.

#### 4.1.2. False Negative Cases

False negative arises when a positive pair that contains two videos presenting the same or very similar motions but our encoder generates two dissimilar representations. Training our model with a large dataset that includes these variations could most likely mitigate this issue. Based on our examination on the false negative cases, almost all false negative cases fall into one of the following three categories.

The first, but an extremely rare category, is due to different ending in the video. A subject could enroll to the verification system with a video that has a few frames of neutral face at the end, but their facial motion video for verification could end at the peak of the facial motion. Both videos contain the same facial motion but with different ending. This difference in video ending could be treated as two different facial motions. Our random trimming augmentation method actually solves this problem. Figure 6 shows an example of slight shift of the facial motion in two videos.

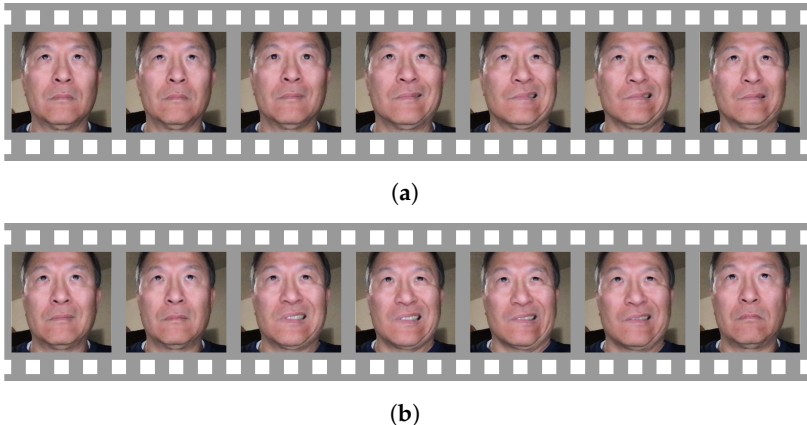

(a)

(b)

**Figure 6.** False negative example with different video ending. (**a**) ends at the peak of the facial motion. (**b**) ends at the neutral face.

The second category is when the subject shows different motion intensity between the enrollment and verification videos. One example of this case is shown in Figure 7. The facial motion in these two videos are the same but with different motion intensity. They should be considered to have the same motion but their representations generated by the encoder could result in a pairwise distance slightly larger than the threshold.

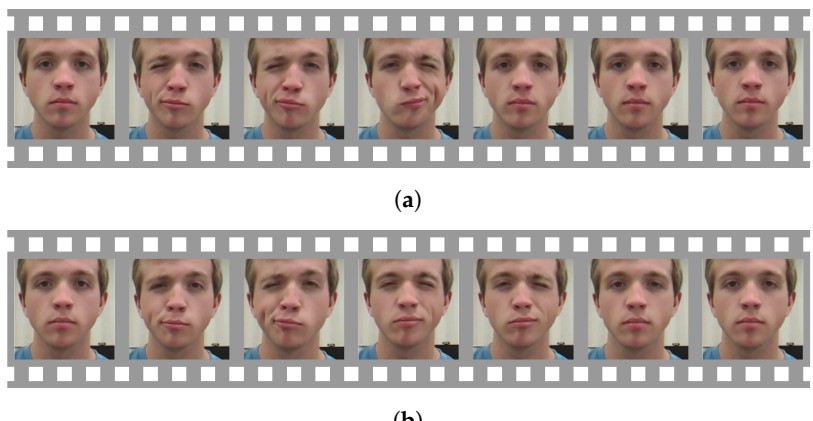

(a)

(b)

**Figure 7.** False negative example with slightly different facial motion intensity. (**a**,**b**) are both the motion pouting lips from right to left, but (**a**) has higher intensity when turning left.

The third category also belongs to inconsistent reiteration of facial motion but with more than intensity change. One example of this case is shown in Figure 8. The difference between these two videos is more than just the facial motion intensity change. The resulting pairwise distance could be much larger than the second category. As a matter of fact, this category should be considered as a borderline case because the difference is too significant to consider them the same motion.

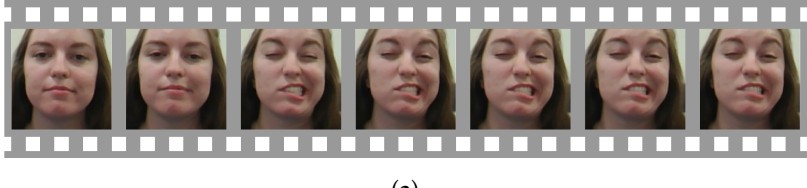

(a)

**Figure 8.** *Cont.*

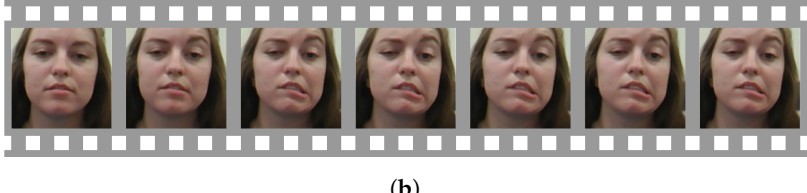

(**b**)

**Figure 8.** False negative example with significantly different facial motion intensity. Subject considers (**a**,**b**) as the same motion. Beside the intensity change on lips, (**b**) also has the left eye always opened.

*4.2. Other Potential Applications*

We run the t-SNE algorithm to visualize the face motion embedding vectors generated by the best model we obtained. The results of five subjects making three to five different motions are shown in Figure 9. It shows that our facial motion encoder is capable of extracting efficient embedding to distinguish different facial motions. These results demonstrate that our approach could generate facial motion embedding as the input to the human-computer interaction system and has great potential for more facial motion based applications.

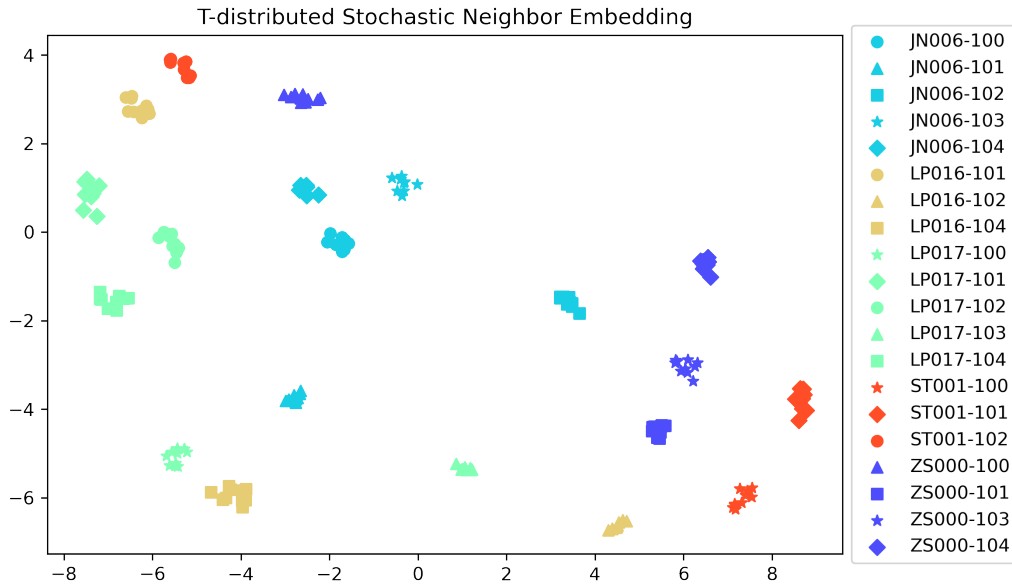

**Figure 9.** t-SNE visualization for the embedding vectors corresponding to three to five customized facial motions from five subjects in our dataset.

*4.3. Future Research Directions*

The spatial feature extractor tuned on facial landmark detection task worked well for our sequence encoder. There are a few improvements could be considered for future work. The FLD model we currently use only predicts 2D coordinates of facial landmark points. Recent research shows that it is feasible to predict 3D face mesh with single RGB image [36,37]. These dense 3D points provide a more accurate description of the face in image. This means its backbone network could retain more spatial features during feed-forwarding. The spatial feature extractor that is pretrained for face mesh prediction tasks could be an even better option for our work.

Both spatial feature extractors adopted in this work were tuned only for a single task. Some research works show that a CNN model could gain boosted performance when jointly trained for multiple tasks [38]. This means that we could potentially develop a more powerful spatial feature extractor with multi-task learning. This approach could require an elaborately designed loss function to ensure the model's success for each subtask. Despite its complexity, it could be a promising optimization to our study.

Our facial motion encoder uses MobileFaceNet, which was derived from MobileNetV2 but optimized for face analysis tasks. Researchers at Apple, Inc. have recently designed a new lightweight architecture called MobileViT [39] that was inspired by the idea of vision transformer. It outperformed MobileNetV2 for both mobile object detection and semantic segmentation tasks. It is reasonable to believe that MobileViT or its variations could boost the performance of our facial motion encoder.

## 5. Conclusions

In this paper, we proposed a unique application of analyzing short facial motion video for identity verification. This is accomplished by the use of a spatial feature extractor followed by a sequence encoder. The spatial feature extractor is a portion of a CNN model, MobileFaceNet, that was originally trained for image-based face analysis. We tune it to be an effective spatial feature extractor for our facial motion analysis application. Our sequence encoder takes the spatial feature extractor output of each frame of the video as its input and identify the positional encoding. The transformer layers will then generate a unique facial motion embedding for the input video. This was accomplished by training with our implementation of a contrastive loss. Due to the uniqueness of our approach, the publicly available datasets do not fully represent the problem that we are solving. Due to this situation we created our own dataset of 59 subjects and a total of 5411 videos.

We used ten-fold cross-validation under varying experimental settings to validate the performance of our method. We achieved a strong average precision of 98.8% on identify verification by using our facial motion analysis. We also discussed the reason behind certain misidentified cases such as flash motions, varying ending of the motion, varying intensity of the motion, and inconsistent reiterations of supposedly the same facial motion. There are various potential applications of this method of identify verification including human-computer interaction. Future work could look into using 3D points, using a CNN that is trained for multiple tasks, and using new lightweight architectures as the pretrained model to tune the spatial feature extractor.

**Author Contributions:** Conceptualization, Z.S. and D.-J.L.; Data curation, A.W.S. and S.A.T.; Formal analysis, Z.S., A.W.S. and S.A.T.; Investigation, D.-J.L.; Methodology, Z.S.; Project administration, D.-J.L.; Resources, D.-J.L.; Software, A.W.S. and S.A.T.; Validation, Z.S.; Writing—original draft, Z.S.; Writing—review & editing, A.W.S., S.A.T. and D.-J.L. All authors have read and agreed to the published version of the manuscript.

**Funding:** This research received no external funding.

**Conflicts of Interest:** The authors declare no conflict of interest.

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
