# Peer review of "Learning Facial Motion Representation with a Lightweight Encoder for Identity Verification"

_electronics, doi:10.3390/electronics11131946_

Round 1

Reviewer 1 Report

The paper presents a method for identity verification relying on facial motion. In particular, the paper focuses on learning facial motion features using a spatial encoder followed by a sequence encoder. The usage of a triplet contrastive loss allows the model to learn discriminative features to represent facial motions tied to identities.

I have a few comments I would like the authors to address:

- The sequence encoder performs a sort of temporal max pooling, which has the advantage of making the descriptor invariant to the number of frames. However pooling is known to remove ordering information, which might be relevant for verification via motion cues. Does this have some impact on the model? I would like the authors to discuss this in the manuscript

- The paper mentions (L111) inference times of milliseconds. On which architecture? Model complexity could be better characterized.

- Literature review should be expanded including works that model temporal cues of adopt motion features related to faces. A few examples:

* Liu, Yong-Jin, et al. "A main directional mean optical flow feature for spontaneous micro-expression recognition." IEEE Transactions on Affective Computing 7.4 (2015): 299-310.

* Becattini, Federico, et al. "PLM-IPE: A Pixel-Landmark Mutual Enhanced Framework for Implicit Preference Estimation." ACM Multimedia Asia. 2021. 1-5.

*Liu, Xin, et al. "Region based parallel hierarchy convolutional neural network for automatic facial nerve paralysis evaluation." IEEE Transactions on Neural Systems and Rehabilitation Engineering 28.10 (2020): 2325-2332. 

- A conclusion section should be added to the manuscript.

Reviewer 2 Report

1. From line 21 - 23 need reference.

2. Figure 1 (face image) please mention the source in the figure title.

3. Line 46 GPU or ASIC need abbreviation.

4. Mention source of the face images in all figure titles.

5. follow same style in all references.
